# HIV gp120 Induces the Release of Proinflammatory, Angiogenic, and Lymphangiogenic Factors from Human Lung Mast Cells

**DOI:** 10.3390/vaccines8020208

**Published:** 2020-05-03

**Authors:** Giancarlo Marone, Francesca Wanda Rossi, Antonio Pecoraro, Valentina Pucino, Gjada Criscuolo, Amato de Paulis, Giuseppe Spadaro, Gianni Marone, Gilda Varricchi

**Affiliations:** 1Department of Public Health, Section of Hygiene, University of Naples Federico II, 80138 Naples, Italy; gcmarone@hotmail.it; 2Azienda Ospedaliera Ospedali dei Colli, Monaldi Hospital Pharmacy, 80131 Naples, Italy; 3Department of Translational Medical Sciences, University of Naples Federico II, 80138 Naples, Italy; francescawanda.rossi@unina.it (F.W.R.); anthonypek@msn.com (A.P.); gjada91@hotmail.it (G.C.); depaulis@unina.it (A.d.P.); spadaro@unina.it (G.S.); 4Center for Basic and Clinical Immunology Research (CISI), 80138 Naples, Italy; 5World Allergy Organization (WAO), Center of Excellence, 80138 Naples, Italy; 6Institute of Inflammation and Ageing, College of Medical and Dental Sciences, University of Birmingham, Birmingham B15 2 TT, UK; valentina.pucino@gmail.com; 7Institute of Experimental Endocrinology and Oncology “G. Salvatore”, National Research Council (CNR), 80131 Naples, Italy

**Keywords:** angiogenesis, histamine, HIV, gp120, IgE, leukotriene C_4_, lymphangiogenesis, mast cells, prostaglandin D_2_, superantigen

## Abstract

Human lung mast cells (HLMCs) express the high-affinity receptor FcεRI for IgE and are involved in chronic pulmonary diseases occurring at high frequency among HIV-infected individuals. Immunoglobulin superantigens bind to the variable regions of either the heavy or light chain of immunoglobulins (Igs). Glycoprotein 120 (gp120) of HIV-1 is a typical immunoglobulin superantigen interacting with the heavy chain, variable 3 (V_H_3) region of human Igs. The present study investigated whether immunoglobulin superantigen gp120 caused the release of different classes of proinflammatory and immunoregulatory mediators from HLMCs. The results show that gp120 from different clades induced the rapid (30 min) release of preformed mediators (histamine and tryptase) from HLMCs. gp120 also caused the de novo synthesis of cysteinyl leukotriene C_4_ (LTC_4_) and prostaglandin D_2_ (PGD_2_) from HLMCs. Incubation (6 h) of HLMC with gp120 induced the release of angiogenic (VEGF-A) and lymphangiogenic (VEGF-C) factors from HLMCs. The activating property of gp120 was mediated through the interaction with IgE V_H_3^+^ bound to FcεRI. Our data indicate that HIV gp120 is a viral superantigen, which induces the release of different proinflammatory, angiogenic, and lymphangiogenic factors from HLMCs. These observations could contribute to understanding, at least in part, the pathophysiology of chronic pulmonary diseases in HIV-infected individuals.

## 1. Introduction

The human immunodeficiency virus (HIV-1) affects more than 36 million people worldwide [Unaids. UNAIDS Data 2018 (2018)]. Although the combined antiretroviral therapy (ART) can successfully suppress HIV viremia and delays the progression of disease [1], the chronic infection requires lifetime treatment due to the viral persistence in latent reservoirs [2,3,4]. Importantly, a significant percentage of HIV-infected individuals has hepatitis C virus (HCV) co-infection [5] resulting in increased HIV reservoir size [6]. The advent of ART has improved survival of HIV-infected adults and children leading to chronic illnesses such as different pulmonary diseases [7,8]. For instance, chronic obstructive pulmonary disease (COPD), asthma, pulmonary hypertension, lung cancer, and asthma are prevalent among HIV patients [7,9,10,11,12,13]. 

Mast cells are immune cells localized in murine [14,15,16] and human lung [17,18,19,20]. Mast cells are critical sentinels in immunity [21,22] and were canonically considered key effectors of allergic responses [18,23,24,25,26]. However, increasing evidences indicate that these cells are involved in bacterial and viral infections [27,28,29,30], pulmonary diseases [22,31,32], angiogenesis [33,34,35,36,37], lymphangiogenesis [38,39], autoimmune disorders [40,41,42], and cancer [43,44,45,46]. There is compelling evidence that human mast cell progenitors can be infected by HIV and retain the virus with maturation in vitro [47,48,49,50]. Importantly, HIV-1 can replicate in latently infected human mast cells [51] and these cells are an inducible reservoir of persistent infection [51,52,53].

Human mast cells display the high-affinity receptor (FcεRI) for immunoglobulin E (IgE) and cross-linking of the IgE-FcεRI network causes the release of preformed (e.g., histamine, tryptase) and de novo synthesized lipid mediators (e.g., prostaglandin D_2_: PGD_2_ and cysteinyl leukotriene C_4_: LTC_4_). Human lung mast cells [33], like macrophages [54,55], basophils [56], and neutrophils [57], also release angiogenic (e.g., vascular endothelial growth factor A: VEGF-A) and/or lymphangiogenic factors (e.g., vascular endothelial growth factor C: VEGF-C) [23,33,55]. Human mast cells isolated from various organs [58,59] are heterogeneous with respect to the mediators produced [19]. 

Different bacteria and viruses synthesize a variety of proteins, termed “superantigens” (SAgs) that activate T and B cells [60,61,62,63,64]. T cell SAgs bind to the MHC class II molecules and to the Vβ domain of the T cell receptor (TCR) and bypass the conventional presentation of antigens by antigen-presenting cells (APCs) [65,66,67,68]. B cell SAgs are endowed with immunoglobulin (Ig)-binding capacity and bind to either the heavy (H)- or light (L)-chain of Igs [69,70,71]. HIV glycoprotein 120 (gp120) is a viral immunoglobulin SAg, because it binds to Igs V_H_3^+^ [72,73,74,75,76,77], the largest of human Ig germline V_H_ family. Therefore, gp120 can stimulate a large percentage of Ig-bearing immune cells, including mast cells. 

Chronic lung diseases, such as chronic obstructive pulmonary disease (COPD), lung cancer, pulmonary hypertension, and asthma, are currently a leading concern for patients with HIV infection [7,8,9,10,11,12,13]. There is compelling evidence that mast cells and their proinflammatory and angiogenic mediators are involved in these disorders [26,32,33,45,78]. In this study we evaluated whether viral gp120 superantigen can induce the release of proinflammatory, angiogenic, and lymphangiogenic factors from primary mast cells isolated from human lung parenchyma.

## 2. Materials and Methods 

### 2.1. Reagents

The following were purchased: bovine serum albumin (BSA), Pipes [piperazine-N,N′-bis (2-ethanesulfonic acid)], L-glutamine, antibiotic-antimycotic solution (10,000 IU penicillin, 10 mg/mL streptomycin, and 25 μg/mL amphotericin B), LTC_4_, and PGD_2_ (Sigma-Aldrich, St. Louis, MO, USA), collagenase (Worthington Biochemical Co., Freehold, NJ, USA), Fetal calf serum (FCS) (GIBCO, Grand Island, NY, USA), and pronase (Calbiochem, La Jolla, CA, USA), RPMI 1640 with 25 mM HEPES buffer, Eagle’s minimum essential medium (Flow Laboratories, Irvine, UK), Percoll (Pharmacia Fine Chemicals, Uppsala, Sweden), (^3^H)-LTC_4_ and (^3^H)-PGD_2_ (New England Nuclear, Boston, MA, USA), CD117 MicroBead (Miltenyi Biotech, Bologna, Italy). The rabbit anti-LTC_4_ and anti-PGD_2_ antibodies and the monoclonal antibody anti-FcεRI were a gift of Dr. Lawrence M. Lichtenstein (The Johns Hopkins University, Baltimore, MD, USA). 

### 2.2. Recombinant HIV gp120 Proteins

Recombinant gp120_MN_, gp120_SF2_, gp120_LAV_, and gp120_CM_ were obtained through the AIDS Research and Reference Reagent Program (National Institute of Allergy and Infectious Diseases, USA) [79,80]. Their characteristics are summarized in Table 1.

### 2.3. Human IgG Anti-IgE

Human IgG anti-IgE (H-aIgE) was purified from the serum of a patient with severe atopic dermatitis as previously described [81,82]. The specificity and activity of IgG anti-IgE were tested as described elsewhere [81].

### 2.4. Human Monoclonal IgM and Human Polyclonal IgG

Monoclonal IgM were purified from the serum of patients with Waldenström’s macroglobulinemia as described elsewhere [83]. Variable regions of these monoclonal IgM were determined using a panel of primary sequence-dependent V_H_ family specific reagents that identify framework regions [84]. Human polyclonal IgG were purified from the serum of healthy donors [85].

### 2.5. Isolation of HLMCs

The study was approved by the Ethics Committee of the University of Naples Federico II (Protocol: Human MC No 7/19, 16/01/2019). The lung tissue was obtained from patients seronegative for HIV-1, HCV, and HBV undergoing thoracic surgery, mostly for lung cancer. HLMC were purified from human lung tissue by a modification of the method previously described [33]. The enzymatic dispersion tissue yields ≈ 5 × 10^5^ mast cells per gram of lung tissue. The purity of these populations ranged from 3% to 18%. HLMCs were partially purified by flotation through a discontinuous Percoll gradient [83]. Mast cell purity using this technique ranged from 47% to 79% and was assessed by alcian blue staining.

### 2.6. Assays of Histamine, LTC_4_, and PGD_2_

HLMCs (≈3 × 10^4^ mast cells per tube) were resuspended in Pipes buffer containing, in addition to Pipes (25 mM), CaCl_2_ (2 mM) and dextrose (1 g/L) and 0.3 mL of the cell suspensions were placed in 12 × 75 mm polyethylene tubes [86]. 0.2 mL of each prewarmed releasing stimulus was added, and incubation was continued at 37 °C for 30 min [87]. Histamine was measured in duplicate determinations with an automated fluorometric technique [88]. LTC_4_ and PGD_2_ were measured in duplicate determinations by radioimmunoassay [87,89]. The anti-LTC_4_ and anti-PGD_2_ antibodies have less than 1% cross-reactivity to other eicosanoids [87,89].

### 2.7. VEGF-A and VEGF-C Release

HLMCs (≈8 × 10^4^ mast cells/per tube) were incubated (37 °C, 6 h) in RPMI 1640 containing 5% FCS, 2 mM L-glutamine, and 1% antibiotic-antimycotic solution, and activated with various concentrations of gp120. At the end of incubation, cells were centrifuged (1000× *g*, 4 °C, 5 min) and the supernatants were stored at −80 °C for subsequent assay of mediator release. VEGF-A and VEGF-C were measured in duplicate determinations using ELISA kits (R&D System, Minneapolis, MN, USA [90]. The ELISA sensitivity is 31.1–2000 pg/mL for VEGF-A and 62–4000 pg/mL for VEGF-C.

### 2.8. Statistical Analysis

Values were expressed as means ± SEM (standard error of the mean). The one-way repeated measures analysis of variance (ANOVA) with Greenhouse–Geisser corrections was used to examine the variations of continuous variables at different experimental conditions. Results were analyzed with GraphPad Prism software (version 8.01; GraphPad Software, La Jolla, CA, USA), and *p* values of less than 0.05 were considered significant.

## 3. Results

### 3.1. Effect of Human IgG Anti-IgE on Mediator Release from HLMCs

IgG anti-IgE (H-aIgE), purified from a small percentage of atopic dermatitis patients, induces histamine and LTC_4_ release from human basophils and mast cells [81]. The activating property of H-aIgE is mediated by the interaction with IgE on basophils and mast cells [82,91]. We used this human autoantibody to activate HLMCs in vitro. H-aIgE (10^−2^ to 3 μg/mL) caused a concentration-dependent histamine secretion from four different preparations of HLMCs isolated from HIV-1-negative subjects (Figure 1). As a control, we used IgG (10^−2^ to 3 μg/mL) purified from four normal donors which did not induce histamine release from HLMCs. These results indicate that mast cells purified from human lung have FcεRI-bound IgE. 

### 3.2. Effects of gp120 from Divergent HIV Isolates from Different Clades on Mediator Release from HLMCs

In a group of experiments we compared the effects of four recombinant gp120 (gp120_MN_, gp120_SF2_, gp120_LAV_, and gp120_CM_) derived from divergent HIV-1 isolates from different viral clades (B and E) of various geographical origins (United States, France, and Thailand) [74] (Table 1) on mediator release from HLMCs. These divergent samples of gp120 concentration-dependently (3 to 60 nM) induced histamine release from HLMCs (Figure 2). These results imply that the capacity to induce mediator release from HLMCs is a general feature of gp120 which has maintained throughout the evolution of the virus.

### 3.3. Correlation between Histamine and Tryptase Release Induced by gp120 from HLMCs

Tryptase is a neutral protease that is a selective marker for mast cells [92,93]. Large quantities of tryptase reside in the secretory granules of all mature human mast cells [92,94]. Activation of HLMCs with gp120 caused the release of tryptase as well as of histamine. Figure 3 shows that there was a positive correlation between the percentage of histamine and tryptase release induced by gp120 (r = 0.66; *p* < 0.001). These data demonstrate that tryptase, contained in secretory granules of mast cells, is released in parallel with histamine, implying that these cells are the source of both mediators found in supernatants of gp120-activated HLMCs.

### 3.4. Effect of Lactic Acid on gp120-Induced Histamine Release from HLMCs

Brief exposure to lactic acid removes IgE bound on FcεRI^+^ cells, thus inhibiting the activating properties of IgE-mediated stimuli [95]. Lactic acid exposure completely blocked the activating effect of both gp120 and by H-aIgE on histamine release from HLMCs (Figure 4). By contrast, the activating property of the mAb cross-linking the α-chain of FcεRI [84] was not modified by this treatment. These results suggest that gp120 induces histamine release from HLMCs through the interaction with IgE bound on mast cells.

### 3.5. Effects of Different IgM Myeloma Proteins on gp120-Induced Mediator Release from HLMCs

To evaluate the mechanism whereby gp120 activates HLMCs, gp120 was incubated with monoclonal IgM of different V_H_ families following the procedure previously described [84]. gp120 (10 nM) was preincubated (15 min, 37 °C) with increasing concentrations (0.1 to 10 μg/ml) of a preparation of monoclonal IgM V_H_3^+^ or monoclonal IgM V_H_6^+^. HLMCs were then added, and the incubation was continued for 30 min at 37 °C. At the end of this incubation, histamine was measured in the supernatants. Preincubation with a preparation of monoclonal IgM which has the V_H_3 domain, concentration-dependently inhibited the effect of gp120 on histamine release from HLMCs (Figure 5). By contrast, a monoclonal IgM which has a V_H_6 domain, had no effect. These results are compatible with the hypothesis that binding of gp120 to the V_H_3 domain of human monoclonal IgM inhibits the interaction with IgE bound to FcεRI on HLMCs.

### 3.6. Effects of gp120 on the De Novo Synthesis of PGD_2_ and LTC_4_ from HLMCs

Activated human mast cells can rapidly synthesize several lipid mediators [23,89]. Prostaglandin D_2_ (PGD_2_) is the main cyclooxygenase metabolite synthesized de novo by human mast cells [78,96]. Activated human mast cells also synthesize cysteinyl leukotriene C_4_ (LTC_4_) through the 5-lipoxygenase pathway [96,97,98]. Both lipid mediators exert a variety of proinflammatory and vasoactive effects [99,100]. In four experiments, we evaluated the production of PGD_2_ and LTC_4_ from HLMCs in response to increasing concentrations of gp120. Figure 6 shows that gp120 (3 to 60 nM) caused the de novo synthesis of both PGD_2_ (Figure 6A) and LTC_4_ (Figure 6B). Figure 6C shows that there was a significant correlation between the production of PGD_2_ and of LTC_4_ caused by gp120 from HLMCs (r =0.55; *p* < 0.001).

### 3.7. Effects of gp120 on the Release of Angiogenic and Lymphangiogenic Factors from HLMCs 

Vascular endothelial growth factors (VEGFs) are essential regulators of the development and functions of blood and lymphatic vessels [101,102]. VEGFs play a major role in new vessel formation and in pulmonary pathophysiology [33,103]. VEGF-A is the most potent proangiogenic molecule [34,104], whereas VEGF-C plays an essential role in inflammatory and tumor lymphangiogenesis [101,102]. Therefore, we investigated the effects of increasing concentrations (10 to 60 nM) of gp120 on the release of angiogenic (VEGF-A) and lymphangiogenic factors (VEGF-C) from HLMCs. HLMCs were cultured (6 h at 37 °C) with gp120 and, at the end of incubation, the release of VEGF-A and VEGF-C was assayed in the supernatants of mast cells. Figure 7 shows that gp120 concentration-dependently caused the release of VEGF-A and VEGF-C from four different preparations of HLMCs.

## 4. Discussion

Primary mast cells isolated from human lung parenchyma of HIV-1 negative subjects can be activated by a human IgG anti-IgE isolated from a patient with atopic dermatitis. These results suggest that HLMCs bind IgE which has a role in allergic diseases [18,105], in pulmonary disorders [22,31] and also in HIV-1 infection [106,107,108,109,110,111]. Viral (gp120) superantigen activates HLMCs to release preformed (histamine and tryptase) and de novo synthesized proinflammatory mediators (LTC_4_ and PGD_2_), angiogenic (VEGF-A), and lymphangiogenic (VEGF-C) factors. The activating property of gp120 appears to be mediated by interaction with IgE V_H_3^+^ on HLMCs. 

Mast cells, present in strategic locations of human lung [19,20,26,32], are involved in several pulmonary diseases [22,31], lung remodeling [112,113], COPD [9,12,32], lung cancer [114,115,116], and asthma [17,19]. In recent years, the widespread use of ART has improved the survival of people with HIV [1], leading to the emergence of chronic inflammatory lung diseases as a noteworthy concern in this population [7,9]. COPD [9,11,12,117], lung cancer [10,118], pulmonary hypertension [13], and bronchial asthma [7,119] occur at high frequency among HIV-infected individuals. HIV infection induces a state of chronic inflammation characterized by persistent immune dysregulation [120], which likely induces the release of proinflammatory cytokines. Although our results were obtained in an experimental model in vitro, the release of several proinflammatory mediators from gp120-activated human lung mast cells may contribute to the pathophysiology of chronic pulmonary diseases in HIV-infected patients.

There is compelling evidence that serum IgE levels are elevated in subjects with HIV infection compared to control [106,107,108,109,110,111], suggesting that mast cells and perhaps other immune cells expressing FcεRI (e.g., dendritic cells, macrophages, basophils) could be involved in various aspects of this infection [27,47,73,91]. Human mast cell progenitors can be infected HIV and retain the virus with their maturation [47,48,49,50]. Moreover, human mast cells are important reservoir of persistent HIV infection [51,52,53]. The involvement of mast cells in HIV infection is not unprecedented. In fact, these cells are involved in several viral infections such as Sendai virus [121], dengue infection [122,123], herpes simplex [124], influenza [125], hantaviruses [126], and cytomegalovirus [127,128]. 

Angiogenesis [129] plays a role in pulmonary pathophysiology [38,130,131]. VEGF-A is a major mediator of angiogenesis and can be produced by several immune cells [33,55,56,104,132,133,134]. To our knowledge, this is the first evidence that gp120 can induce the release of angiogenic factors from HLMCs raising the possibility that these cells can contribute to angiogenesis, a process of pivotal relevance in lung pathophysiology [38,130,131]. Further studies are needed to comprehensively define the contributive role of lung mast cells to angiogenesis in pulmonary diseases (e.g., COPD, asthma, etc.) [7,9,10,11,12,13] associated with HIV infection.

The mammalian lung is rich of lymphatic vessels [135] which are increased in human lung following infections [136,137,138,139]. We provide the first evidence that a superantigenic viral activation of HLMCs leads to the production of VEGF-C, a major mediator of lymphangiogenesis [140]. Lymphangiogenesis is canonically considered pivotal for the diffusion of metastasis to draining lymph nodes [101,102]. However, recent evidences indicate that VEGF-C can potentially exert protective effects, since inflammation-associated lymphangiogenesis can improve the resolution of inflammation [141,142]. Therefore, the contribution of lung mast cells to lymphangiogenesis during HIV infection commands additional investigations.

## 5. Conclusions

We have previously shown that gp120 causes cytokines (IL-4 and IL-13) release from human basophils [73]. Together with the data of the present study, it is possible to conclude that gp120 [143] can function as a viral superantigen activating HLMCs and basophils to release proinflammatory mediators (histamine, tryptase, PGD_2_, LTC_4_), cytokines (IL-4 and IL-13), and angiogenic/lymphangiogenic factors (VEGF-A and VEGF-C). The latter results could contribute to immune dysfunction in patients with HIV.

The successful rollout of anti-viral therapy ensured that HIV infection is managed as a chronic condition [2,3,4,98]. Persistent inflammation and immune dysregulation associated with HIV lead to accelerated aging and pulmonary diseases [7,8,9,10,11,12,13]. HIV-positive persons are, therefore, exhibiting increasing pulmonary complications. Our results, indicating that gp120 can induce the release of potent proinflammatory mediators (histamine, tryptase, PGD_2_ and LTC_4_) [144,145,146,147,148,149,150] from HLMCs might explain, at least in part, how HIV can cause lung damage.

Our study has a limitation that should be pointed out. The in vitro experiments were performed using primary human lung mast cells purified from HIV^-^ patients undergoing thoracic surgery for lung cancer. Several independent investigators have demonstrated that human mast cells can be infected by HIV [47,48,49,52]. Importantly, human mast cells represent a long-lived reservoir of persistent HIV infection even in patients treated with ART [52,53]. There is also some evidence that circulating levels of histamine, a major mast cell mediator [17], are increased in HIV infected patients [151]. Future studies should compare the activating property of gp120 on lung mast cells isolated from HIV^-^ and HIV^+^ subjects to support the clinical significance of our findings.

In conclusion, our results indicate that immunoglobulin superantigen gp120 can interact with IgE V_H_3^+^ bound to FcεRI to induce the release of proinflammatory, angiogenic, and lymphangiogenic mediators from human lung mast cells. Future studies are needed to investigate whether these observations in vitro can help to understand the pathophysiology of chronic lung diseases among HIV patients.

## Figures and Tables

**Figure 1 vaccines-08-00208-f001:**
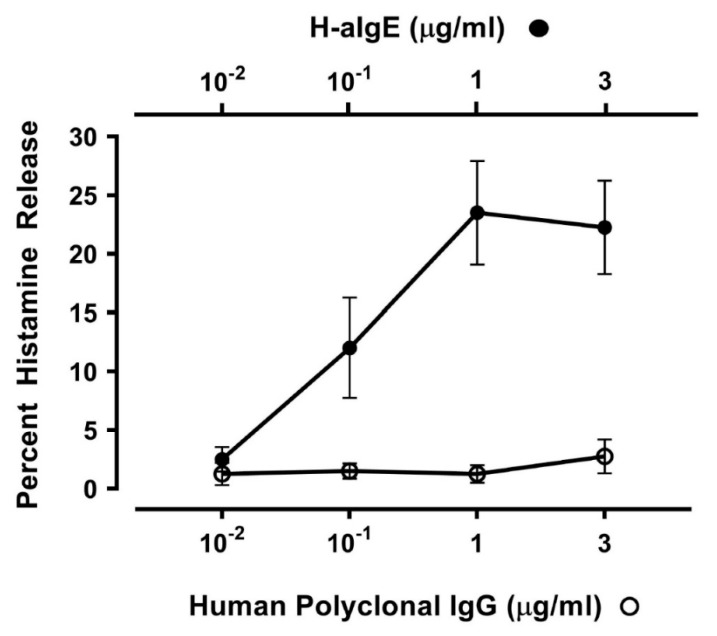
Effects of increasing concentrations of human IgG anti-IgE (H-aIgE) [81] and four preparations of human polyclonal IgG purified from normal donors on histamine release from HLMCs obtained from four donors negative for HIV-1 antibodies. HLMCs were incubated (30 min at 37 °C) with the indicated concentrations of H-aIgE or polyclonal IgG. Each point shows the mean ± SEM obtained from four different experiments.

**Figure 2 vaccines-08-00208-f002:**
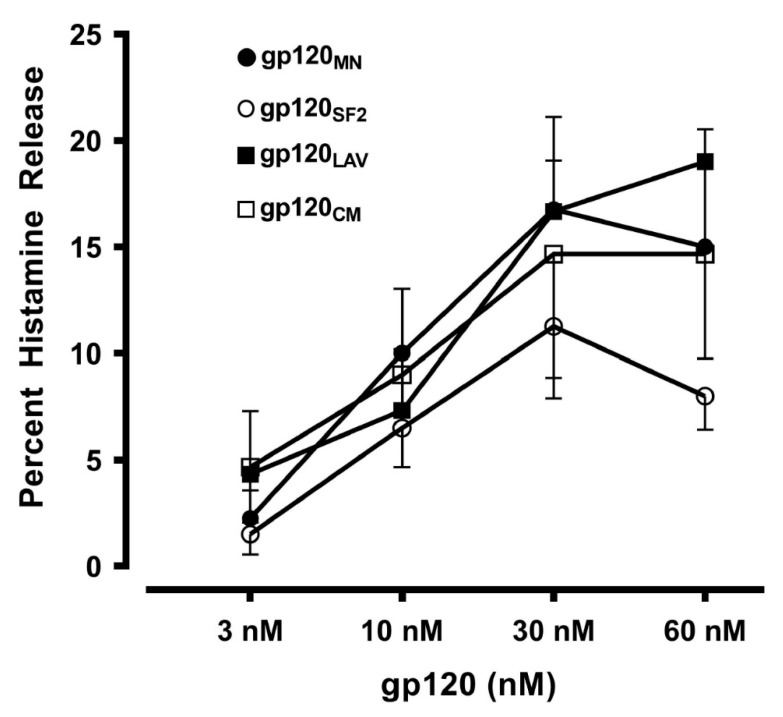
Effects of increasing concentrations of gp120 from four different isolates (gp120_MN_, gp120_SF2_, gp120_LAV_, gp120_CM_) on histamine secretion from HLMCs obtained from four different preparations of HLMCs obtained from donors negative for HIV-1 antibodies. HLMCs were incubated (30 min at 37 °C) with the indicated concentrations of gp120. Each point shows the mean ± SEM obtained from four different experiments.

**Figure 3 vaccines-08-00208-f003:**
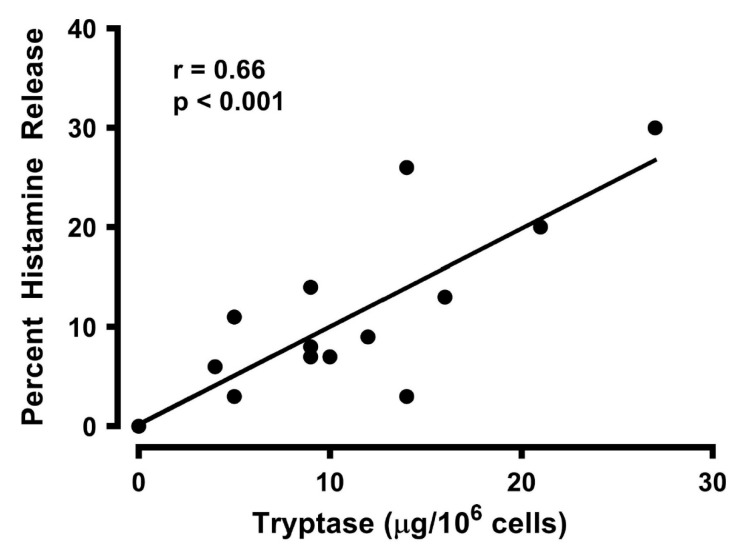
Correlation between the percent histamine release and tryptase secretion caused by gp120 from HLMCs. Each point represents the mean of duplicate determinations from separate experiments. r = 0.66; *p* < 0.001.

**Figure 4 vaccines-08-00208-f004:**
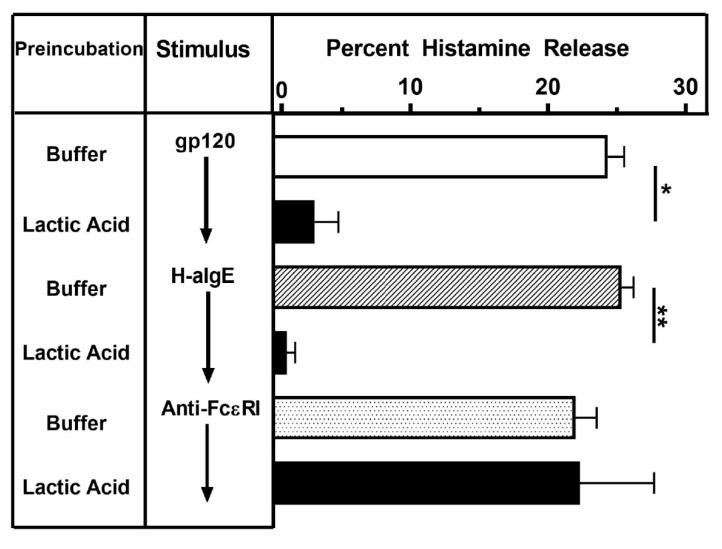
Effects of lactic acid on histamine release from HLMCs induced by gp120, H-aIgE or anti-FcεRI. HLMCs were either treated with buffer or lactic acid (0.01 M, pH 3.9, 5 min at 22 °C) and washed twice. HLMCs were then challenged (30 min at 37 °C) with gp120 (10 nM), H-aIgE (1 μg/mL), or anti-FcεRI (1 μg/mL). Each bar represents the mean ± SEM of histamine release from triplicate incubations. * *p* < 0.05 when compared to cells treated with buffer; ** *p* < 0.01 when compared to cells treated with buffer.

**Figure 5 vaccines-08-00208-f005:**
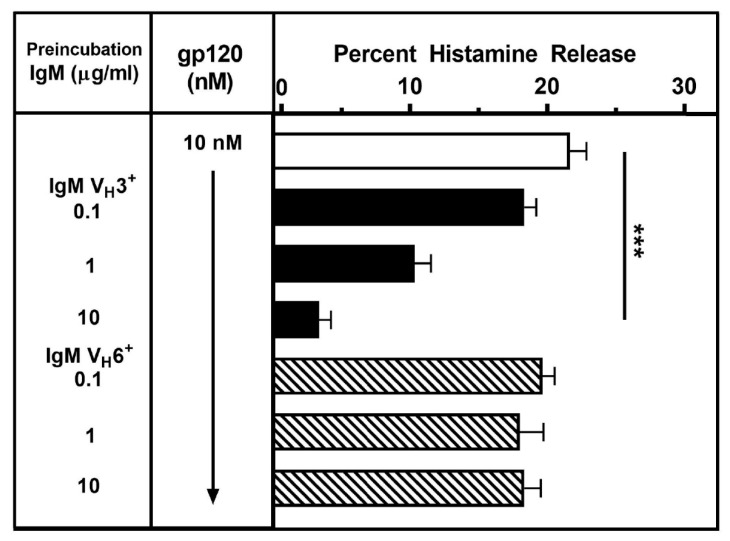
Effects of human monoclonal IgMs on the activation of HLMCs induced by gp120. gp120 (10 nM) was preincubated (15 min at 37 °C) with increasing concentrations (1 to 10 μg/mL) of human monoclonal IgM V_H_3^+^ or IgM V_H_6^+^. HLMCs were then added and incubation continued for 30 min at 37 °C. Each bar shows the mean ± SEM of histamine release from triplicate incubations. *** *p* < 0.001 when compared to controls.

**Figure 6 vaccines-08-00208-f006:**
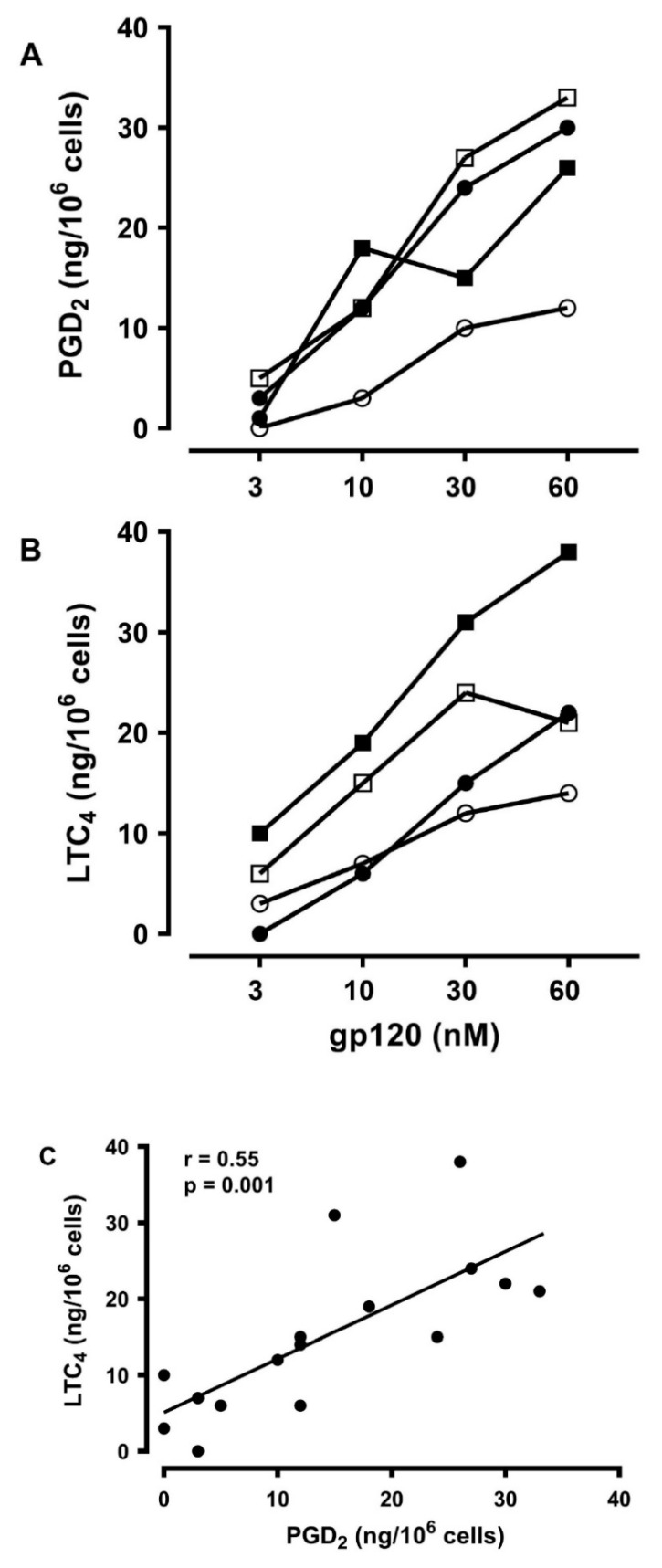
Effects of increasing concentrations of gp120 on the de novo synthesis of PGD_2_ (**A**) and LTC_4_ (**B**) from four different preparations of HLMCs. HLMCs were incubated (30 min at 37 °C) with the indicated concentrations of gp120. Each point is the mean of duplicate determinations (**C**). Correlation between PGD_2_ and LTC_4_ production caused by gp120 from HLMCs. r = 0.55; *p* < 0.001.

**Figure 7 vaccines-08-00208-f007:**
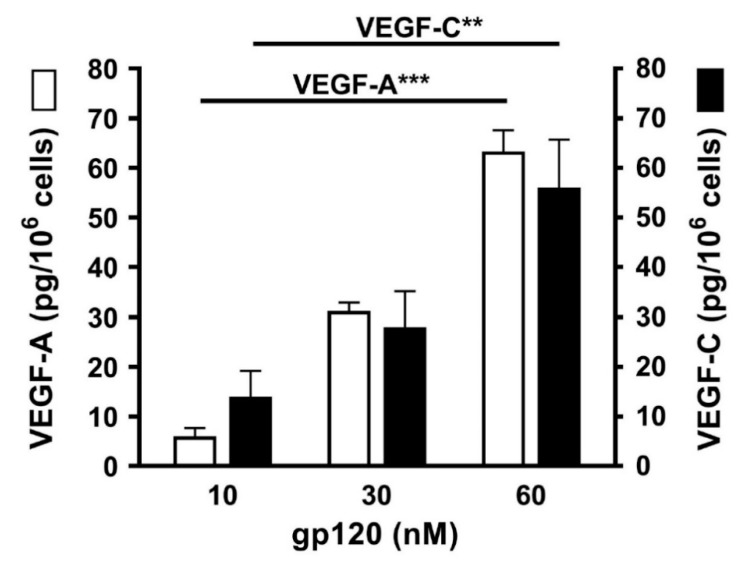
Effects of increasing concentrations of gp120 on the release of VEGF-A and VEGF-C from HLMCs from four different preparations of HLMCs. HLMCs were incubated (6 h at 37 °C) in the presence of the indicated concentrations of gp120. Each bar is the mean ± SEM obtained from four different experiments. ** *p* < 0.01; *** *p* < 0.001.

**Table 1 vaccines-08-00208-t001:** Recombinant HIV gp120 used in this study.

Recombinant Envelope Protein	gp120 Isolate	Clade	Geographic Origin	Expression System
gp120_MN_	MN	B	USA	Insect Cells
gp120_SF2_	SF2	B	USA	CHO Cells
gp120_LAV_	LAV	B	France	Insect Cells
gp120_CM_	CM	E	Thailand	Insect Cells

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
