# Peer review of "HIV gp120 Induces the Release of Proinflammatory, Angiogenic, and Lymphangiogenic Factors from Human Lung Mast Cells"

_vaccines, 2020, doi:10.3390/vaccines8020208_

Round 1
Reviewer 1 Report
The manuscript entitled "HIV gp120 Induces the Release of Proinflammatory, Angiogenic, and Lymphangiogenic Factors from Human Lung Mast Cells," by Marone et al., describes a study investigating whether HIV gp120 can stimulated human mast cells (harvested from HIV- individuals) to release preformed and de novo synthesized proteins via IgE bound to FCξRI.
Overall, the study is well written and the research design appears appropriate. There are several minor spelling and grammar errors that need to be corrected prior to presentation.
While manuscript can be shortened in several places (e.g. Materials and Methods section: the "reagents" and "buffers" sections seem superfluous and the information can be integrated into the experimental methods), it would be useful to provide additional information to illustrate why mast cell activation by gp120 has importance in clinical disease. For example, are the differences in the pathology of asthma or COPD between HIV+ and HIV- individuals? If yes, what are they and how are they associated with mast cell activation? While the authors clearly demonstrate that gp120 is able to activate pulmonary mast cells in vitro, the significance of this finding remains unclear.
More importantly, the authors indicate that HIV infects and is maintained in human mast cell lineages. This begs the question of whether HIV infection has an effect on the activation and/or mediator release of the infected mast cell (i.e. will findings of this study be applicable to mast cell responses from infected individuals?) If HIV infection alters the mast cell response, the results of this present study may be irrelevant in the study of pulmonary pathology in HIV+ individuals. Therefore, addition of a control, consisting of HIV infected mast cells, would be warranted.
Author Response
The manuscript entitled "HIV gp120 Induces the Release of Proinflammatory, Angiogenic, and Lymphangiogenic Factors from Human Lung Mast Cells," by Marone et al., describes a study investigating whether HIV gp120 can stimulated human mast cells (harvested from HIV- individuals) to release preformed and de novo synthesized proteins via IgE bound to FCξRI.
Overall, the study is well written and the research design appears appropriate. There are several minor spelling and grammar errors that need to be corrected prior to presentation.
We thank the Reviewer for her/his positive comments and constructive criticisms.
While manuscript can be shortened in several places (e.g. Materials and Methods section: the "reagents" and "buffers" sections seem superfluous and the information can be integrated into the experimental methods), it would be useful to provide additional information to illustrate why mast cell activation by gp120 has importance in clinical disease. For example, are the differences in the pathology of asthma or COPD between HIV+ and HIV- individuals? If yes, what are they and how are they associated with mast cell activation? While the authors clearly demonstrate that gp120 is able to activate pulmonary mast cells in vitro, the significance of this finding remains unclear.
We have corrected the grammar and spelling errors.
We agree with the Reviewer and we have shortened the Materials and Methods section.
The Reviewer raised an important point. Although our results in vitro are straightforward, their clinical significance remains unclear. As suggested by the Reviewer, we have emphasized the rationale of our experiments in the Introduction (lines 104-109) and, more importantly, we have discussed their possible clinical relevance (Discussion, lines 241-251).
More importantly, the authors indicate that HIV infects and is maintained in human mast cell lineages. This begs the question of whether HIV infection has an effect on the activation and/or mediator release of the infected mast cell (i.e. will findings of this study be applicable to mast cell responses from infected individuals?) If HIV infection alters the mast cell response, the results of this present study may be irrelevant in the study of pulmonary pathology in HIV+ individuals. Therefore, addition of a control, consisting of HIV infected mast cells, would be warranted.
We agree with the Reviewer that it would be important to compare the activating property of gp120 on lung mast cells isolated from HIV- and HIV+ patients. A large number of donors are needed to perform the complex in vitro experiments presented in this study. The number of HIV+ patients undergoing thoracic surgery for lung cancer is rather limited. Therefore, it would be extremely difficult to repeat the experiments presented in this paper using lung mast cells from HIV+ patients. However, we have discussed the importance of this point raised by the Reviewer in the revised Discussion (lines 287-294).
Reviewer 2 Report
The manuscript by Marone et al. reports new in vitro data on the release of different classes of mediators from HLMCs induced by immunoglobulin superantigen gp120. The authors showed that the HIV gp120 viral superantigen interaction with IgE VH3+ as a primary target in HLMCs isolated from human patients can be a primary mechanism of this induction. The manuscript is well structured and all methods are adequately described. I also agree with all conclusions made by the authors. The manuscript may only require some edits to fix grammar and improve writing. Some commas are missing, in some cases the meaning of the sentence is vague. For example:
"Collectively, our data indicate that HIV gp120 is a viral superantigen which can interact with IgE VH3+ to induce the release of different proinflammatory, angiogenic, and lymphangiogenic factors from HLMCs". Comma is missing after superantigen, the verb "can" here does not provide the precise meaning of the major discoveries of the manuscript.
Author Response
The manuscript by Marone et al. reports new in vitro data on the release of different classes of mediators from HLMCs induced by immunoglobulin superantigen gp120. The authors showed that the HIV gp120 viral superantigen interaction with IgE VH3+ as a primary target in HLMCs isolated from human patients can be a primary mechanism of this induction. The manuscript is well structured and all methods are adequately described. I also agree with all conclusions made by the authors. The manuscript may only require some edits to fix grammar and improve writing. Some commas are missing, in some cases the meaning of the sentence is vague. For example:
"Collectively, our data indicate that HIV gp120 is a viral superantigen which can interact with IgE VH3+ to induce the release of different proinflammatory, angiogenic, and lymphangiogenic factors from HLMCs". Comma is missing after superantigen, the verb "can" here does not provide the precise meaning of the major discoveries of the manuscript.
We thank the Reviewer for her/his positive comments.
We have fixed the grammar and improved the English.
We agree with the Reviewer; we have eliminated “can” and added “comma after superantigen” (Abstract, line 56).
Reviewer 3 Report
vaccines-778920-peer-review-v1
HIV gp120 Induces the Release of Proinflammatory, 2 Angiogenic, and Lymphangiogenic Factors from 3 Human Lung Mast Cells
Abstract
- I think the structure of the abstract needs to show the background information, methodology and analysis, results and conclusion
- I was trying to look the ‘study design’ and I cannot see it. The authors need to clearly structure so as that the reader will grasp easily
- I can see in the main document that you applied One Way Anova but again make clear this in the appropriate place
- The ultimate goal of a research, either basic or applied, is to benefit the public. Could the authors add a single statement what the HIV gp120 research contributes to: early HIV detection, ART (antiretroviral therapy) discovery, or virological assessment follow up
Introduction
- Line 76: I am not sure about the number of people affected by HIV in 2018— it’s nearly 36 million so please revisit this statement
- Line 76: HCV? Hepatitis C virus? Write in full.
- I don’t understand what were the previous gaps on this issue, and which gap this research fills?
- The authors need to revise what previous research studies found and what their gaps were, so that they will show the gap they filled
Methodology
- I invited other laboratory experts to review this methodology
Results
- None
Discussion
- I think in each finding; the clinical and public health relevance should be added— I can see some of the findings are linked in this way but the great majority of them are not linked to the clinical and public health significance.
Conclusion
- Could this be a separate heading?
Author Response
Comments and Suggestions for Authors
HIV gp120 Induces the Release of Proinflammatory, 2 Angiogenic, and Lymphangiogenic Factors from 3 Human Lung Mast Cells
Abstract
- I think the structure of the abstract needs to show the background information, methodology and analysis, results and conclusion
We agree with the Reviewer and we have modified the structure of the Abstract following the style of the journal.
- I was trying to look the ‘study design’ and I cannot see it. The authors need to clearly structure so as that the reader will grasp easily.
The purpose of our study is indicated in lines 48-50.
- I can see in the main document that you applied One Way Anova but again make clear this in the appropriate place.
The statistical analysis is described in details in “Materials and Methods” (section 2.8).
- The ultimate goal of a research, either basic or applied, is to benefit the public. Could the authors add a single statement what the HIV gp120 research contributes to: early HIV detection, ART (antiretroviral therapy) discovery, or virological assessment follow up.
We have added a sentence suggesting that our results could contribute to understand, at least in part, the pathophysiology of chronic pulmonary diseases in HIV-infected subjects (lines 57- 59).
Introduction
- Line 76: I am not sure about the number of people affected by HIV in 2018— it’s nearly 36 million so please revisit this statement.
The Reviewer is absolutely right. “26” was a typographical mistake. We have corrected the number of HIV affected people worldwide (line 72).
- Line 76: HCV? Hepatitis C virus? Write in full.
We have written HCV in full (line 76).
- I don’t understand what were the previous gaps on this issue, and which gap this research fills?
- The authors need to revise what previous research studies found and what their gaps were, so that they will show the gap they filled
This study demonstrates, to our knowledge, for the first time, that HIV gp120 is a viral superantigen which interacts with IgE VH3+ to induce the release of different classes of mediators from human lung mast cells. We recognize that our in vitro results cannot be immediately translated to clinical relevance. We have added a paragraph (Discussion section, lines 241-251) to illustrate the background, the novelty and the possible clinical significance of our findings.
Methodology
- I invited other laboratory experts to review this methodology
The techniques used in this study have been used in our Laboratory for several years.
Results
- None
Discussion
- I think in each finding; the clinical and public health relevance should be added— I can see some of the findings are linked in this way but the great majority of them are not linked to the clinical and public health significance.
We agree with the Reviewer and we have modified the Discussion section. First, we have added a paragraph to explain how our results could contribute to understand, at least in part, the pathophysiology of chronic pulmonary diseases occurring at high frequency among HIV-infected individuals (lines 241-251). Second, we have recognized the limitation of our in vitro study and the need of additional studies to support the clinical significance of our findings (lines 287-294).
Conclusion
- Could this be a separate heading?
We prefer to mantain the Conclusions with the Discussion section.
Round 2
Reviewer 1 Report
The manuscript entitled "HIV gp120 Induces the Release of Proinflammatory, Angiogenic, and Lymphangiogenic Factors from Human Lung Mast Cells" by Marone et al. has been resubmitted for review following revision. The authors addressed the majority of the comments provided by the reviewer. While there are still some areas of concern regarding applicability of results to clinical disease, the current manuscript is acceptable for publication.